

# Dense water formation in the Eastern Mediterranean under global warming scenario

Iván M. Parras-Berrocal[1], Rubén Vázquez[1,2], William Cabos[2], Dimitry V. Sein[3,4], Oscar Álvarez[1], Miguel Bruno[1], Alfredo Izquierdo[1]

[1]Instituto Universitario de Investigación Marina (INMAR), Universidad de Cádiz, Puerto Real, Cádiz 11510, Spain
[2]Department of Physics and Mathematics, University of Alcalá, Alcalá de Henares 28801, Spain
[3]Alfred Wegener Institute for Polar and Marine Research, Bremerhaven 27570, Germany
[4]Shirshov Institute of Oceanology, Russian Academy of Science, Moscow 117997, Russia

*Correspondence to*: Iván M. Parras-Berrocal (ivan.parras@uca.es)

**Abstract.** Dense water formation in the Eastern Mediterranean (EMed) is essential in sustaining the Mediterranean overturning circulation. Changes in the sources of dense water in the EMed point to changes in the circulation and the water properties of the Mediterranean Sea. Here we examine with a regional climate system model the changes in the dense water formation in the EMed through the twenty-first century under the RCP8.5 emission scenario. Our results show a shift in the dominant source of Eastern Mediterranean Deep Water (EMDW) from the Adriatic Sea to the Aegean Sea at the first half of twenty-first century. The projected dense water formation reduces by 75% for the Adriatic Sea, 84% for the Aegean Sea and 83% for the Levantine Sea by the end of the century. The reduction in the intensity of deep water formation is related to hydrographic changes of surface and intermediate water, that strengthen the vertical stratification hampering the vertical mixing and thus the convection. Those changes have an impact on the water that flows through the Sicilian Strait to the Western Mediterranean and therefore on the whole Mediterranean system.

**1 Introduction**

The Eastern Mediterranean Sea (EMed) is a key region where intermediate and deep water convection is regularly observed leading to a vertical recirculation (Roether et al. 1996), which is essential in sustaining the Mediterranean thermohaline circulation (MTHC). The Atlantic Water (AW) flows through the Strait of Gibraltar compensating the Mediterranean Sea freshwater deficit (Bethoux and Gentili 1999; Sanchez-Gomez et al. 2011), getting denser through its path to the EMed, 25   becoming the Modified Atlantic Water (MAW). Intermediate water in the Levantine Sea is formed by wintertime air-sea interactions that cool the MAW increasing its density originating the Levantine Intermediate Water (LIW; LIWEX group, 2003; Millot et al. 2014). The LIW spreads westward at intermediate depths (150-600 m), and then to the Atlantic Ocean driving the main thermohaline circulation cell of the Mediterranean (Lascaratos et al., 1993; Vargas-Yáñez et al., 2012; Millot et al., 2019). Along its path through the eastern and western Mediterranean, the warmer and saltier LIW preconditions



the surface waters for deep water formation (DWF) in regions such us the Gulf of Lions, the Adriatic, or the Aegean Seas during the winter months (MEDOC Group, 1970).

The Adriatic Sea has been identified as the main source of the Eastern Mediterranean Deep Water (EMDW), as it fills the EMed deep layers (Pollak, 1951; Malanotte-Rizzoli et al., 1997). In winter, the Adriatic DWF is triggered by (i) the cold and dry Bora winds (north-easterly) which induce large surface buoyancy loss as a result of a rapid surface cooling and strong

evaporation (Lascaratos et al., 1999) and (ii) the presence of LIW which favours the deepening of the convective layer (Mantziafou and Lascaratos, 2008). Here, most of the DWF takes place in southern Adriatic through open convection inside the Southern Adriatic Pit depression where it is strongly preconditioned by the presence of a permanent cyclonic gyre (Lascaratos et al., 1999; Manca et al., 2002; Mantziafou and Lascaratos, 2008). Moreover, a smaller amount of deep water is also formed on the continental shelf of the northern and middle Adriatic. During 1990s, hydrographic surveys showed that

the EMDW was mostly formed in the Aegean Sea inducing to the so-called Eastern Mediterranean Transient (EMT; Roether et al., 1996). The main mechanisms leading to the DWF in the Aegean Sea are the open sea convection due to the cooling of surface water, the preconditioning of cyclonic circulation and the basin salinity increase (Nittis et al., 2013). The main formation sites are the Cyclades plateau, the Syros-Chios basins, and the Creta Sea. Nittis et al. (2013) point out that the annual DWF rate for the Aegean Sea (0.24 Sv) during the EMT was comparable to the 0.3 Sv formed each year in the

Adriatic Sea for the 1979-1994 period. Consequently, the Adriatic Sea and the Aegean Sea compete to be the dominant source of DWF in the EMed (Roether et al., 2014). The leading role depends on the water density conditions reached during winter months (Klein et al., 2000).

According to projections of future climate, the Mediterranean Sea has been identified as one of the most responsive regions to climate change. By the end of the 21st century the IPCC scenarios project a warmer and dryer Mediterranean climate (Ali

et al., 2022). The sea surface temperature (SST) is expected to increase from 0.5 to 3.7ºC, while intermediate and deep layers warm by 0.8-3.0ºC and 0.15-0.18ºC, respectively (Somot et al., 2006, 2008; Adloff et al., 2015; Darmaraki et al; 2019; Parras-Berrocal et al., 2020; Soto-Navarro et al., 2020; Ali et al., 2022). These changes under global warming may impact on the main mechanisms leading to the DWF and on the MTHC. A recent study has analysed the future response of the main spots for DWF in the Mediterranean to climate change in downscaled climate simulations, pointing to a reduction of deep

convection in all regions (Soto-Navarro et al., 2020). In fact, the DWF in the north-western Mediterranean is projected to collapse by mid-21st century due to the increases in the vertical density gradient between surface and intermediate waters, which strengths the stratification in the water column, hampering the deep convection (Parras-Berrocal et al., 2022).

The dense (intermediate and deep) water formation in the EMed has been extensively studied (e.g., Roether et al., 1996; Lascaratos et el., 1999; Nittis et al., 2003; Mantziafou and Lascaratos, 2008; Androulidakis et al., 2012; Dunic et al., 2018;

Li and Tanhua, 2020). A lot of attention has been also focused on the causes of the EMT and its impacts, especially on the Mediterranean circulation (e.g., Roether et al., 1996, 2014; Borzelli et al., 2009; Beuvier et al., 2010; Incarbona et al., 2016). However, the response of the EMed dense water formation to climate change has only been briefly assessed by Somot et al. (2006), Adloff et al. (2015) and Soto-Navarro et al. (2020), so a detailed analysis about the expected change and their causes



is needed. Thus, the aim of this work is to study the impact of climate change on the dense water formation in the EMed
(Adriatic, Aegean and Levantine Sea, Figure 1) by the end of the century, as well as to identify the mechanisms involved in
those changes. To address this issue we use a regional climate system model (RCSM), which has been widely employed to
analyse the present and future climate of the Mediterranean Sea (Darmaraki et al., 2019; Parras-Berrocal et al., 2020; de la
Vara et al., 2022), including the interannual variability of DWF at the north-western Mediterranean (Parras-Berrocal et al.,
2022).

The paper is organized as follows: In Sect. 2 the RCSM and the simulations employed in this work are described. In Sect. 3,
the present-climate and future evolution of intermediate and deep water formation in the EMed are analysed. Finally, the
discussion of the results and the conclusions are contained in Sect. 4.

## 2 RCSM setup and simulations

In this work we use the RCSM ROM (REMO-OASIS-MPIOM; Sein et al. 2015). In ROM, the regional atmosphere model
REMO (Jacob et al., 2001) is coupled to the global oceanic model MPIOM (Max Planck Institute Ocean Model; Jungclaus et
al., 2013; Marsland et al., 2003) via the OASIS3 coupler (Valcke, 2013). ROM also comprises other sub-models such as the
HAMburg Ocean Carbon Cycle model (Maier-Reimer et al., 2005), the Hydrological Discharge model (Hagemann and
Dümenil-Gates, 1998, 2001), a soil model (Rechid and Jacob, 2006) and a dynamic/thermodynamic sea ice model (Hibler,
1979) which are treated as modules either of the atmosphere or the ocean.

REMO is formulated on a rotated grid with the center of the domain situated around the Equator in the rotated coordinates
with a constant horizontal resolution of 25 km and a total of 27 hybrid levels. The atmospheric domain used in this study
extends to the North Atlantic, the eastern tropical Pacific and the Mediterranean Sea (de la Vara et al., 2022; Vazquez et al.,
2022). MPIOM has an orthogonal curvilinear grid with a variable horizontal resolution ranging from 7 km at the south
Alboran Sea to 25 km in the eastern coasts of the Levantine Sea. The horizontal resolution in the Adriatic and the Aegean
seas is not coarser than 16 and 19 km, respectively. On the vertical, the model has 40 z-levels with increasing layer thickness
with depth, from 16 m at the surface to 550 m near the seafloor (Parras-Berrocal et al., 2020). Contrary to other RCSMs
developed to analyse the Mediterranean Sea, the water exchange at Gibraltar is not parametrized and the properties of the
Atlantic waters are not relaxed toward climatological values. In fact, the water exchange through the Strait of Gibraltar is
explicitly reproduced which allows to propagate the North Atlantic signal into the Mediterranean Sea, and vice-versa. The
spin-up of MPIOM was performed conforming to Sein et al. (2015). First, MPIOM runs in stand-alone mode starting with
climatological temperature and salinity data (Levitus et al., 1998). Afterwards, it is integrated four times through the 1958–
2002 period forced by ERA40. For the coupled runs, the model starts from the final state reached in the last stand-alone run
and integrated again, forced two times by ERA40 and one time by ERA-Interim reanalysis (1979–2012). Then, it runs for 56
years (1950–2005) starting from the last state of the coupled simulation forced by ERA-Interim. More information about the



model parametrization and setup, as well as a detailed evaluation of Mediterranean present climate and future changes
simulated by ROM can be found in Parras-Berrocal et al. (2020).

To assess the ROM performance reproducing the dense water formation in the EMed, we use a simulation forced by ERA-
Interim (1980-2012) which has been previously defined as ROM_P0 in Parras-Berrocal et al. (2020, 2022). In order to study
the interannual variability of dense water formation in the EMed during the present and future climate we take the data from
1976 to 2005 period of the historical run (ROM_P1) and from 2006 to 2099 of the climate change projection under the
RCP8.5 scenario (ROM_P2). All simulations employed in this work are part of the Med-CORDEX initiative
([www.medcordex.eu](www.medcordex.eu)).

## 3 Results

### 3.1 Present-day EMed Deep Water Convection

In this section, we evaluate the ROM_P0 skills reproducing the average rate and the interannual variability of DWF in the
EMed during the present climate (1980-2012). To quantify the DWF over the Adriatic, Aegean, and Levantine Seas we have
computed the annual DWF rate for a specific isopycnic surface ($\sigma_\theta$). The DWF rate is calculated from the difference
between the maximum volume of water denser than $\sigma_\theta$ for a given year minus the minimum volume of that water for the
previous year (Somot et al., 2018). According to those authors, following the volume of the deep water for a given $\sigma_\theta$
(denser than $\sigma_\theta$) is the best quantitative way to study the DWF in a model output.

The interannual DWF rate in the Adriatic Sea (Figure 2a) agrees well (r=0.70) with estimates based on the Princeton Ocean
Model (POM) of Mantziafou and Lascaratos (2008). ROM_P0 (POM) shows annual rates ranging from 0.01 (0.12) Sv to
0.50 (0.93) Sv. During 1981-1999, ROM_P0 produces 5.45 Sv yr of newly waters denser than 29.0 kg/m³ corresponding to
an annual formation rate of 0.29 Sv (Figure 2a), which is comparable to the 0.3 Sv estimated by Roether and Schlitzer (1991)
from tracer data. Mantziafou and Lascaratos (2008) simulated 7.51 Sv yr of total volume of deep water ($\sigma >29.1$ kg/m³),
corresponding to an annual rate of 0.40 Sv, which overestimates the mean annual rate of Roether and Schlitzer (1991). On
the other hand, the interannual DWF rate in the Aegean Sea (Figure 2b) is also well correlated (r=0.75) with the POM results
reported by Nittis et al. (2003). The total volume of deep water formed during 1981-1994 by ROM_P0 ($\sigma >28.95$ kg/m³)
corresponding to an annual formation rate of 0.30 Sv. This is close to values presented in Nittis et al. (2003), where the total
volume formed at the same period is 3.62 Sv yr ($\sigma >29.2$ kg/m³) leading to an annual rate of 0.26 Sv (Figure 2b).

ROM_P0 has demonstrated a good performance simulating the average and interannual DWF rate in the Adriatic and
Aegean Seas. However, the potential density of the newly formed waters in ROM_P0 is lower than those presented by
observations (Figure S1) and other models. We find that the potential density of ROM_P0 at 650 m depth are 0.1 kg/m³ and
0.2 kg/m³ lighter than WOA18 (Boyer et al., 2018) for the Adriatic and Aegean (Figure S1), respectively. The deep water
generated in the Adriatic Sea has densities between 29.1 kg/m³ and 29.25 kg/m³ (Schlitzer et al. 1991; Gacic et al., 2002;
Mantziafou and Lascaratos, 2008) while in the Aegean Sea is denser than 29.2 kg/m³ (Klein et al., 1999; Nittis et al., 2003;




Beuvier et al., 2010). In ROM_P0 the DWF takes place at σ >29.0 kg/m³ for the Adriatic Sea and at σ >28.95 kg/m³ for the Aegean (Figure 2).

In the Levantine basin, for ROM_P0, the total volume of intermediate and deep water (σ >28.7 kg/m³) formed during the period 1981-2012 is 22.3 Sv yr, which correspond to a mean yearly production of 0.69 Sv. This amount of water produced in the Levantine Sea agrees very well with the 0.69 Sv suggested by Lascaratos (1993), calculated for water denser than 28.92 kg/m³ with a mixed-layer model. The formation rate simulated by ROM_P0 is consistent with previous estimates that range between 0.6 and 1.3 Sv (Ovchinnikov, 1984; Tziperman and Speer, 1994; Lascaratos et al., 1999). As well as for the Adriatic and Aegean Sea, the potential density simulated by ROM_P0 at 300 m depth in Levantine Sea is 0.15 kg/m³ lighter than WOA18 (Figure S1). The potential density of the intermediate and deep water formed in the Levantine Sea by ROM_P0 σ >28.7 kg/m³ is lower than the σ >28.85 kg/m³ defined by Lascaratos (1993). Despite the lighter densities, we have identified the isopycnals in which the DWF occurs in ROM_P0 for the studied areas. This allows us to assess the projected climate change signal in the EMed DWF under the RCP8.5 scenario with the aim to contribute to the Med-CORDEX initiative generating useful climatic information.

**3.2 Impact of climate change on the dense water formation at the EMed**

We now examine the yearly evolution of DWF rate in the twenty first century over the main spots for dense water formation in the EMed. To achieve this aim, we have computed the DWF rate for the isopycnals that we have identified in the previous section as the isopycnals where the DWF takes place in ROM model. Thus, we have used 29.0 kg/m³ as lower density bound for the Adriatic Sea, 28.95 kg/m³ for the Aegean Sea and 28.7 kg/m³ for the Levantine Sea.

In the Adriatic, Aegean, and Levantine Seas the averaged DWF rates for 1976-2005 are 0.32±0.09 Sv, 0.25±0.22 Sv and 0.80±0.30 Sv, while for 2070-2099 under RCP8.5 are 0.08±0.04 Sv, 0.04±0.02 Sv and 0.14±0.18 Sv (Figures 3a, 3b and 3c), respectively. The DWF rate is expected to decrease by 75% in the Adriatic, 84% in the Aegean and almost by 83% in Levantine Sea for the 2070-2099 period compared to 1976-2005. The results suggest that the reduction of the dense water formation in the three studied regions starts by mid-21st century under the RCP8.5 scenario. During the 2005-2040 period the total volume of deep water formed in the Aegean Sea (13.4 Sv yr) is higher than in Adriatic (9.93 Sv yr) (Figures 3a, 3b), which means a shift in the dominant source of EMDW.

In order to identify the mechanisms leading to the projected reduction of dense water formation in the EMed, we have evaluated (i) the role of the winter air-sea fluxes through the accumulated surface buoyancy loss (BL) and (ii) the stratification index (SI). The BL (Equation 2) was computed as the time integral of the buoyancy flux (BF, Equation 1) for every year (Y) of the 1976-2099 from December of the previous year (T1) to March (T2) and averaged over the Adriatic, Aegean and Levantine basins. The BF is calculated as the sum of contributions of heat and freshwater fluxes (Marshall and Schott, 1999; Somot et al. 2018; Parras-Berrocal et al. 2022):

$$BF = g \cdot \left( \frac{\alpha \cdot Q_{net}}{\rho_0 \cdot c_p} + \beta \cdot SSS \cdot FWF \right), \qquad (1)$$



$$BL(Y) = -\int_{T_1}^{T_2} BF \cdot dt \, , \tag{2}$$

where $Q_{net}$ and FWF are the net surface heat and freshwater fluxes, respectively (both positive downward), g is the gravitational acceleration (9.81 ms$^{-2}$), α and β the thermal expansion and haline contraction coefficients (respectively calculated as a function of surface T and S), $\rho_0$ the reference density of sea water 1025 kg/m$^3$, Cp the specific heat capacity of sea water (equal to 4000 Jkg$^{-1}$ºK$^{-1}$) and SSS the sea surface salinity.

To assess the pre-winter water column stratification we have computed the SI using December data. Low values of SI

indicate a weak stratification in the water column. The SI is often used in Mediterranean studies (Somot et al., 2018; Margirier et al., 2020, Parras-Berrocal et al., 2022) and it is calculated following Turner (1973).

$$SI = \int_0^h N^2 \, zdz, \tag{3}$$

where N is the Brunt-Väisälä frequency ($N^2 = g/\rho_0 \, \partial_\rho/\partial_z$), z is the depth, ρ the potential density and h the maximum depth of integration which we have chosen to be 650 m because it is right below the LIW layer (150-600 m; Menna and

Poulain, 2010).

Our results indicates that the intensity of DWF rate is mostly determined by the SI (Pearson correlation coefficient (r) > 0.7 in all regions), as low or high amount of water produced can be found with similar BL values (r < 0.1) (Figure 3). For the 1976-2099 period the BL (SI) has a mean value of 0.81±0.13 m$^2$s$^{-2}$ (1.17±0.31 m$^2$s$^{-2}$), 0.89±0.18 m$^2$s$^{-2}$ (1.68±0.38 m$^2$s$^{-2}$) and 0.88±0.14 m$^2$s$^{-2}$ (1.73±0.46 m$^2$s$^{-2}$) for the Adriatic, Aegean and Levantine Seas, in that order. Higher values of DWF rate are

linked to a low SI, which reflects a weaker stratification of the water column (Figure 3a, 3b, 3c). The BL does not show changes in the interannual variability neither in the trend. However, from 2040s the SI shows a change in the trend in the three regions, especially remarkable at the Aegean and Levantine basins. From 2040, the SI steadily increase in the Adriatic (5.43·10$^{-3}$ m$^2$s$^{-2}$y$^{-1}$) showing a maxima of 1.86 m$^2$s$^{-2}$. The SI rise sharply in the Aegean (9.59·10$^{-3}$ m$^2$s$^{-2}$y$^{-1}$) and even more in the Levantine (1.52·10$^{-2}$ m$^2$s$^{-2}$y$^{-1}$) reaching values close to 2.5 m$^2$s$^{-2}$ in both regions at the end of 21st century. These changes

strengthen the vertical stratification in the Adriatic, Aegean and Levantine Seas hampering the vertical mixing and thus the deep convection.

The projected increase in SI comes from changes in the hydrographic characteristics of water column. Under the RCP8.5 emission scenario the temperature and the salinity are projected to increase through the whole water column in the main spots for dense water formation in the EMed (Figure 4). By the end of the 21st century the Adriatic Sea will be warmer and

saltier. The temperature in the upper ocean (0-100 m) will increase by 3.3ºC while in the intermediate (100-600 m) and 600-1000 m layers warm by 2.6ºC and 2ºC (Figure 4a). The salinity is expected to increase by: 0.5 psu at 0-100 m and 0.4 psu at 100-600 m and 600-1000 m (Figure 4b). In the Adriatic, the warming and the salinization are practically homogeneous from the surface to deep layers; however in the Aegean and Levantine basins, the temperature and salinity increases start at the surface and are progressively transferred to deeper layers over the years (Figure 4).





The Aegean Sea experience a warming of 3.9ºC in the upper ocean. In the 100-600 m layer the temperature will increase by

2.7ºC while in the deeper layer the warming is lower than 0.8ºC (Figure 4c). The salinity in the upper layer is fresher than in

deeper layers due to the net inflow of lighter water through the Dardanelles strait. It is precisely in this layer where the larger

salinity increase is found (1 psu) by the end of the 21st century (Figure 4d). The 100-600 m and 600-1000 m layers also tend

to get saltier by up to 0.6 and 0.2 psu, respectively. From the surface to intermediate depths the warming and the salinization

accelerate in the second half of the century.

The Levantine basin is where the higher EMed temperatures are found (Figure 4e). The temperature is expected to increase,

in average, by 3.6ºC in the 0-100 m layer. In the 100-600 m and 600-1000 m the temperature will increase by 2.1 ºC and

0.6ºC, which correspond to the lowest warming at those depths in comparison to the other spots for deep convection in the

EMed. Finally, the salinity increases by 0.4 psu in the upper and intermediate layers and by 0.2 psu in the deeper layer by the

end of the century (Figure 4f). As well as in the Aegean the projected increase in temperature and salinity accelerates in the

second half of the century.

To quantify the relative contribution of surface and intermediate water to the reduction in the intensity of DWF, we use the

methodology applied in Parras-Berrocal et al. (2022). We compare the SI calculated from spatially and temporally averaged

vertical profiles in the Adriatic, Aegean and Levantine Seas in four cases (Table 1 and Figures S2, S3 and S4): (a) using the

values corresponding to the historical period (1976-2005, Hist.); (b) using the last thirty years of RCP8.5 projection (2070-

2099, Proj.); and also generating two synthetic profiles, (c) one including Hist. features for the first 100 m depth and Proj.

characteristics for deeper layers (100-650 m depth), and (d) a second one using Proj. properties for the first 100 m depth and

Hist. for deeper layers. In the Adriatic Sea, the results suggest that the change in AdSW characteristics provokes the 50% of

the total SI future change while the contribution of LIW causes the other 50%. In the Aegean Sea, the alteration in

BSW/LSW properties leads the 80.8% of the total SI future change while the change in CIW/LIW is 19.2%. Finally, in the

Levantine Sea changes in the MAW/LSW properties contributes in a 60% whereas LIW in a 40%.

## 4 Discussion and Conclusions

The impact of climate change on dense water formation in the EMed has briefly been assessed in previous studies (Somot et

al., 2006; Adloff et al., 2015; Soto-Navarro et al., 2020). All authors considered the maximum of the winter mixed layer

depth (MLD) as a proxy for deep water convection. The results reported in those works do not show a consensus concerning

the changes in the deep water formation role in the EMed. Somot et al. (2006) found that the maximum MLD decreases by

about 20% for the Aegean Sea and 60% for the Levantine Sea in 2099 under the A2 scenario, whereas no significant change

is expected for the Adriatic Sea. For this region, their simulations projected an increase in the DWF rate; however the ADW

outflowing through the Otranto Strait is lighter which leads to a weakening in the Eastern Mediterranean thermohaline cell.

The results obtained by Adloff et al. (2015) under the A2 scenario by the end of the 21st century, point out to the reduction

of MLD in the Adriatic and Levantine Seas whereas it increases in the Aegean Sea. More recently, Soto-Navarro et al.



(2020) found in a multimodel analysis that most models agree in projecting a reduction in the intensity of DWF in the Adriatic Sea by the end of the century under RCP4.5 and RCP8.5 scenarios. However, in the Aegean Sea the results are not robust among models, as some of them point to a reduction and others to an increase.

Due to the large spread in the simulated magnitude of the changes in the DWF in previous works (Somot et al., 2006; Adloff et al., 2015; Soto-Navarro et al., 2020), we try to identify the mechanisms involved in the expected changes in the EMed DWF using a single model runs; the same used in Parras-Berrocal et al. (2022) for identifying the mechanisms of the future change in DWF in the north-western Mediterranean. The projection was carried out with the RCSM ROM under the high-emission RCP8.5 scenario and it was also a member (AWI25-MPI-8.5) of the ensemble used in Soto-Navarro et al. (2020).

As well as in Parras-Berrocal et al. (2022), the drawback of using a single model runs is that it does not allow us to provide robust conclusions and to generalize our results. However, the advantage of this approach is that it makes it easier to find physically consistent mechanisms responsible for these changes, making a valuable contribution to the Med-CORDEX. Moreover, this kind of works provide essential information for futures studies using multi-model ensemble analysis by deepening in processes. Indeed, the recent work of Simon and Schroeder (2023) highlights the importance and the need of

further research using climate models to understand the mechanisms involved in regional processes such as the Mediterranean dense water formation and its consequences.

    In this work we quantify the deep water convection through the DWF rate. We estimate the annual DWF rate following the volume of deep water for a specific isopycnic surface ($\sigma_\theta$) which is probably the most quantitative way to estimate the DWF, especially in model analysis (Somot et al., 2018). We show that ROM_P0 is capable to reproduce the average and

interannual DWF rates in the main spots for deep convection in the EMed. Moreover, ROM_P0 captures the main features of the EMT (Roether et al., 1996), simulating higher DWF rates in the Aegean Sea than in the Adriatic Sea for winters from 1988 to 1994 (Figure S5). In ROM_P0 the potential density of newly deep water formed in the Adriatic ($\sigma > 29$ kg/m$^3$), Aegean ($\sigma > 28.95$ kg/m$^3$) and Levantine ($\sigma > 28.7$ kg/m$^3$) Seas are lighter than the values presented in the literature (Lascaratos, 1993; Mantziafou and Lascaratos, 2008; Nittis et al., 2003). This could be explained by the negative salinity

bias displayed throughout the entire water column (0-1000 m depth) by ROM_P0 over the EMed (Figure S1). The largest salinity bias is found at the upper layers of the Aegean Sea (Figure S1) as ROM_P0 overestimates the inflow of water through the Strait of Dardanelles, as previously reported by Parras-Berrocal et al. (2020). However, we are confident that this limitation does not impact in our DWF study, as ROM_P0 has demonstrated a good representation of the deep water volume formed during present climate, independently of the lower density values.

Our results project a DWF rate reduction of 75% for the Adriatic Sea, 84% for the Aegean Sea and 83% for the Levantine Sea by the end of the century under the RCP8.5 scenario. Analysing the mechanisms involved in the decrease we observe that strong or weak DWF rates can occur with similar BL values (Figure S6). We find out that changes in the hydrographic properties of the upper and intermediate water masses lead to a higher stratification of the water column at Adriatic, Aegean, and Levantine basins, which hamper the deep convection. Simon and Schroeder (2023) found using a multidecadal (1951-

2020) observation-based analysis that the weakening heat loss is a potential factor of the reduction in dense water formation.





The authors point to a possible dependence between the weakening heat loss and the stronger stratification which remains unexplored. Thus, it is essential for Med-CORDEX community to clarify the relative contribution of both processes in forthcoming works.

In the future, the temperature and salinity show an increase which is especially noticeable in the 0-100 m and 100-600 m layers. Those alterations in temperature and salinity will increase the vertical density gradient, which in turn strengthen the stratification of the water column (Table 1). Recently, Amitai et al. (2021) found that a DWF decrease in the Adriatic Sea creates a state where warmer and saltier intermediate water reaching the north-western Mediterranean significantly affects the deep water convection in the Gulf of Lions. The reduction of EMed DWF as well as the DWF collapse expected in the north-western Mediterranean (Soto-Navarro et al., 2020; Parras-Berrocal et al., 2022) may have an impact on the deep ventilation and on the MTHC. Those changes are reflected on the flows exchanged between the Atlantic Ocean and the Mediterranean Sea through the Strait of Gibraltar (Parras-Berrocal et al., 2022).

As shown in Figure 3, from 2020 to 2040 the DWF rate shows maxima in all DWF spots. During these 20-years, the SI shows minima due to an increase of potential density at 0-100 m layer (Figure S7). In turn, such increased potential density is caused by a salinization of that layer (Figures 4 and S7), which preconditions the convective regions reducing the vertical stratification and leading to the formation of higher volumes of deep water. The increase of the salt content in the 0-100 m layer could be the result of a change in the upper ocean circulation that alters the AW path. During that period, 2020-2040, the anticyclonic gyre located off Libya (Sidra Gyre [33º-35º N; 14º-16º E], in agreement with Menna et al., 2019) is enhanced (Figures S7); this structure retains part of the AW (fresher) weakening its inflow into the Levantine Sea. As a consequence, the circulation (anticyclonic) anomaly is also characterized by negative salinity and density anomalies. This alteration leads to an increase of the upper layer salinity in the Levantine, Aegean, and Adriatic Seas. From 2040 on, the anticyclonic structure returns to its previous state enabling the inflow of AW into the Levantine basin. However, for the second half of the 21st century the density (0-100 m) of EMed decreases (Figure S7). This is induced by the accelerated warming and salinization expected in the upper layer under the RCP8.5 scenario.

Another result worth discussing is that from 2005 to 2040 the annual DWF rate for the Aegean Sea is higher than for the Adriatic Sea in most of the years. For this period, the accumulated deep water in the Aegean Sea is 13.4 Sv yr whereas in the Adriatic Sea it is 9.93 Sv yr. The results suggest a shift in the main source of EMDW from the Adriatic Sea to the Aegean Sea, as previously happened during the EMT. The HadCM2-SUL climate experiment (Thorpe and Bigg, 2000) has also shown a decrease in the intensity of deep convection in the Adriatic during 2040-2060 while it increases in the eastern basins. This is also supported by Adloff et el. (2015) who displayed that the future MTHC tends to be similar to an EMT situation, with Aegean Sea becoming the main source of EMDW in the future.

**Data availability**

The model data are available online (https://doi.org/10.5281/zenodo.7594313).

**Author contributions**

IMP-B, AI and WC planned the study. AI and IMP-B designed the analysis framework. IMP-B processed data conducted the analysis and wrote the manuscript. DS performed the ROM runs. RV, WC, DS, OA, MB, and AI contributed with the analysis performance and interpretation of the results. AI revised and edited the final version of the manuscript. IMP-B prepared everything.

**Competing interest**

The authors declare that they have no conflict of interest.

**Acknowledgements**

Simulations were done at the German Climate Computing Center (DKRZ). This work is part of the Med-CORDEX (www.medcordex.eu) initiative and HyMex program (www.hymex.org).

**Financial support**

I. M. Parras-Berrocal was supported by the Spanish National Research Plan through project TRUCO (RTI2018-100865-B-C22) and the Plan Propio UCA 2022-23. Dmitry V. Sein received funding from the Federal Ministry of Education and Research of Germany (BMBF) in the framework of ACE (grant no. 01LP2004A) and the Ministry of Science and Higher Education of Russia (theme no. FMWE-2021-0014).

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

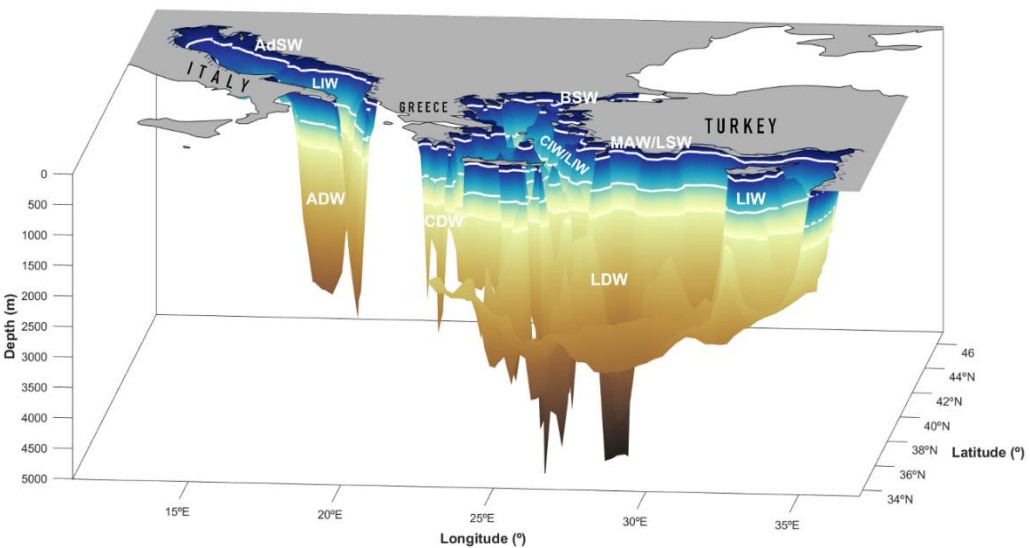

**Figure 1. Bathymetry of the main spots for dense water formation in the EMed: Adriatic Sea, Aegean Sea and Levantine Sea. The main water masses of each spot sorted by depth range are also shown: [0-100 m] Adriatic Surface Water (AdSW), Black Sea Water (BSW), Levantine Surface Water (LSW); [100-650 m] Levantine Intermediate Water (LIW), Cretan Intermediate Water (CIW); [650-1000 m] Adriatic Deep Water (ADW), Cretan Deep Water (CDW) and Levantine Deep Water (LDW).**





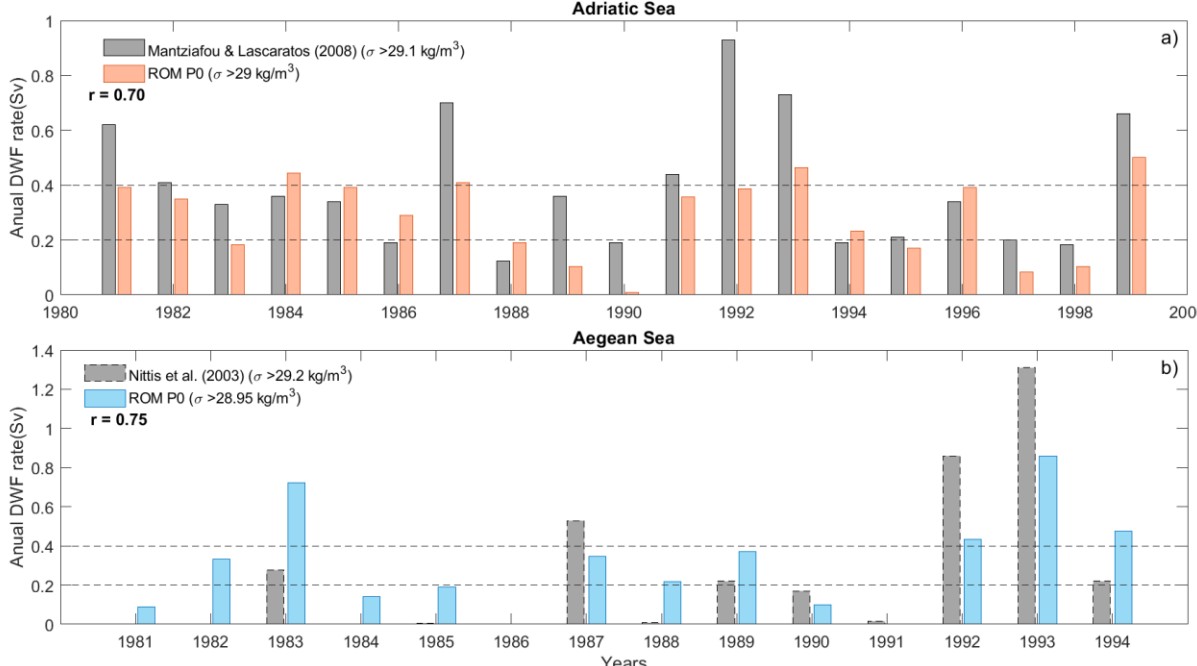

**Figure 2. Annual formation rate (Sv) of deep water in the (a) Adriatic Sea and (b) Aegean Sea. Note different ranges in vertical and horizontal axes.**

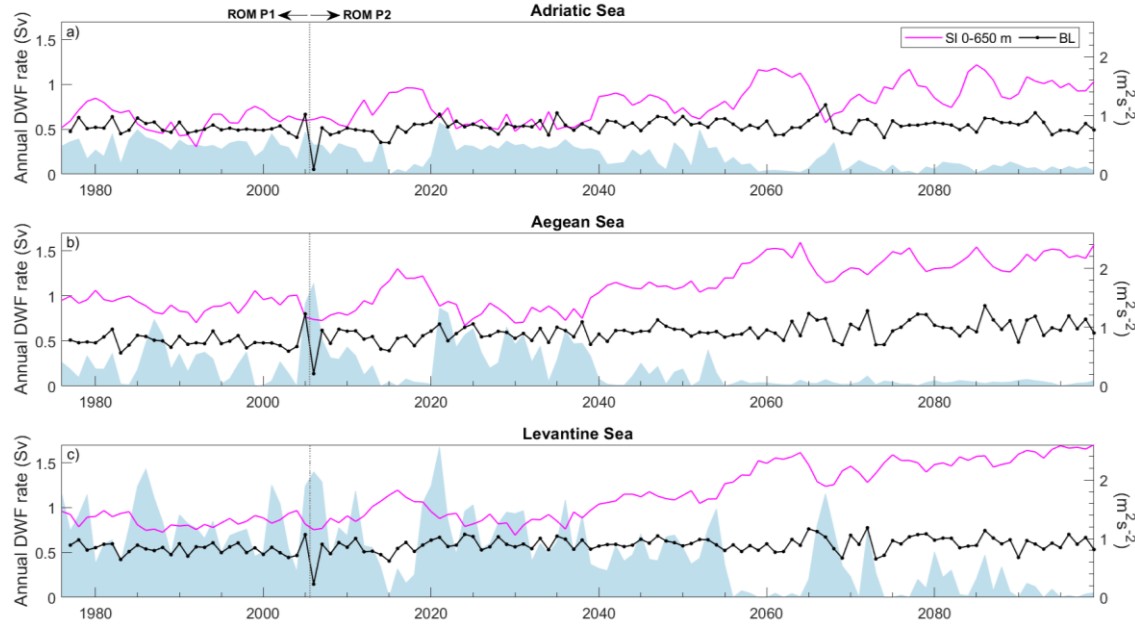

**Figure 3. Time series (1976–2099) of ROM_P1 and ROM_P2 simulations of yearly DWF rate (Sv) (filled area), winter integrated buoyancy loss (BL, m²s⁻²) (doted-black) and stratification index (SI, m²s⁻²) for 0–650 m (magenta) averaged over (a) Adriatic Sea, (b) Aegean Sea and (c) Levantine Sea. All-time series correspond to winter months (December-January-February-March) whereas SI was computed in December of preceding year.**



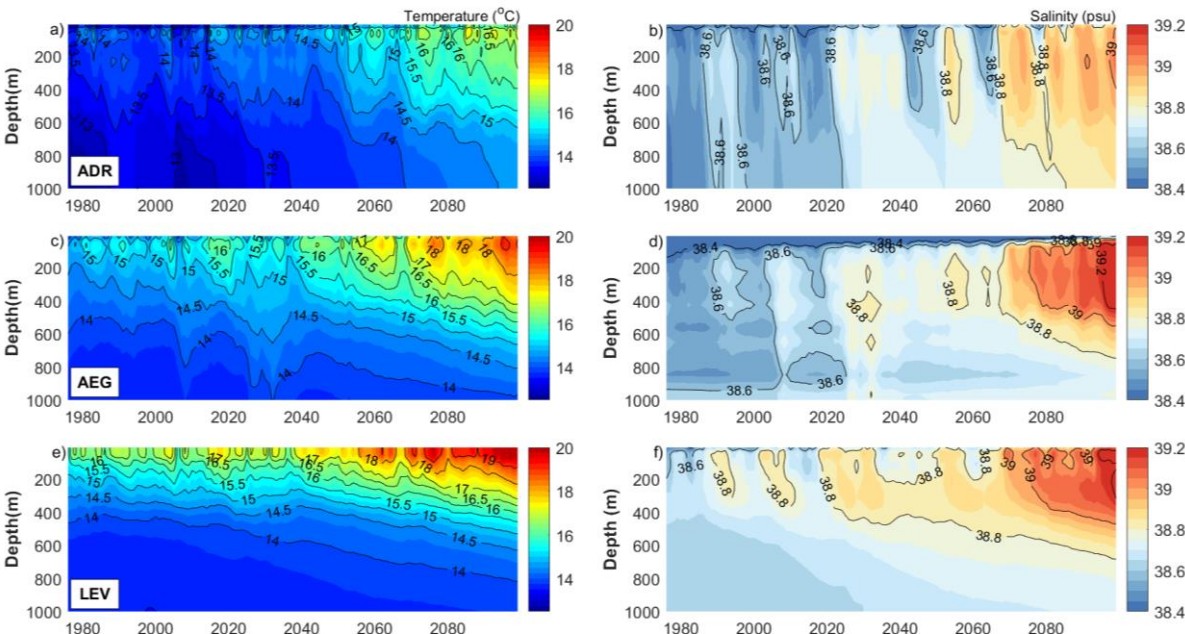

**Figure 4. ROM RCP8.5 time series (1976-2099) of 0-1000 m (a, c, e) potential temperature (ºC) and (b, d, f) salinity (psu). All-time series correspond to winter months (December-January-February-March) in the Adriatic (upper row), Aegean (middle row) and Levantine (bottom row) Seas.**

470

**Table 1. Stratification Index (m²s⁻²) and the quantification of the percentage of surface and intermediate water contributions calculated from vertical profiles presented in Figure S2, S3 and S4.**

|  | Adriatic Sea | | Aegean Sea | | Levantine Sea | |
|---|---|---|---|---|---|---|
|  | *SI* | *%* | *SI* | *%* | *SI* | *%* |
| **Hist. (1976-2005)** | 0.94 | - | 1.40 | - | 1.30 | - |
| **Proj. (2070-2099)** | 1.48 | - | 2.13 | - | 2.35 | - |
| **(0-100 m)<sub>Hist.</sub> + (100-650 m)<sub>Proj.</sub>** | 1.21 | 50.0% | 1.99 | 80.8% | 1.93 | 60.0% |
| **(0-100 m)<sub>Proj.</sub> + (100-650 m)<sub>Hist.</sub>** | 1.21 | 50.0% | 1.54 | 19.2% | 1.72 | 40.0% |