# Peer review of "Dense water formation in the Eastern Mediterranean under global warming scenario"

_EGUsphere, 2023_

## Referee Comment (RC2)

The authors investigated the variations of dense water formation (DWF) in the Eastern Mediterranean (EMed) through the twenty-first century under the RCP8.5 emission scenario for understanding the impacts of climate changes on the Mediterranean overturning circulation. Their results indicated that the dominant source of Eastern Mediterranean Deep Water (EMDM) shifts from the Adriatic Sea to the Aegean Sea during the 2005-2040 period. By the end of the century, DWF for the Adriatic Sea, the Aegean Sean, and the Levantine Sea all perform a pronounced decrease by 75%, 84%, and 83%, respectively, which is a result of hydrographic changes of surface and intermediate water and the associated strengthening water column stratification under the RCP8.5 emission scenario. The results shown are impressive and, as was pointed out in the manuscript, also fill in the gap of the DWF study in the EMed providing a more quantitative assessment than previous studies. The manuscript was also well-written and easy to follow. But some improvements may be needed before the publication.

1) The coverages of the Adriatic Sea, the Aegean, and the Levantine Sea should be specified and shown in a figure as results and discussions of this study focus on the DWF from these regions. Thus, it is important to provide the spatial extent of these basins, which can also help readers to understand the studied area better. In addition, as it was stated that the horizontal resolution of the model varies from 7 km to 25 km (which is a big difference, I think), it is better to show the computational grid as well.

2) Statistics analysis and parameters are needed. Firstly, the authors may need to provide $p$ values for every correlation coefficient as they are important to illustrate the significance. Secondly, the 2040s was regarded as a time point around which sharp changes in SI and DWF (Figure 3) were observed. However, the author may need to provide a more convincing way to address this time point not just by the naked eye but using some statistical tools, like the non-parametric change-point Pettitt test (Pettitt, A.N. A non-parametric approach to the change-point problem. Appl. Stat. 1979, 28, 126–135).

3) Could you double-check the unit "Sv yr" which first appears on Line 113? If my understanding of the unit "Sv" is correct, Line 113 should be rewritten as:

*During 1981-1999, ROM_P0 produces a total of 5.45 Sv of newly waters denser than 29.0 kg/m3 corresponding to an annual formation rate of 0.29 Sv…*

4) Lines 202-211. Although the authors provided descriptions of SI for different periods, I am still not quite sure how the authors calculated the percentage contributions of different water bodies to the temporal changes in SI. Could you please provide some descriptions or equations to further address the calculation?

5) Lines 268-270. It may be a jump to conclude that the increasing potential density is caused by the increasing salinity over the upper 100 m, as the authors only compared the salinity changes and density changes (Figure S7) but ignored the contributions of temperature changes. As shown in Figure 4 subsurface (0-100 m) temperature seemly performs an increase in the ADR (Figure 4a) from the period of 2006-2020 to the period of 2020-2040 but fluctuates in the AEG (Figure 4c) and LEV (Figure 4e). Thus, the author may need to quantify both contributions of the changes in temperature and salinity to the changes in density.

6) Lines 270-271. The authors may need to provide more evidence in addressing the causes of the changes in the upper ocean circulation, like correlations between changes in salinity or temperature and changes in circulation patterns. Or to provide some mechanistic explanations on how the changes in salinity or temperature would lead to changes in circulation. Or to provide results of previous studies here that may have such discussions.

---

## Author Comment (AC1)

We thank the Reviewer #1 for the effort in reviewing the manuscript and for her/his positive evaluation. The posted comments have helped us to improve the manuscript and make it more robust and complete.

**Reviewer #1 (Comments to Author (shown to authors):**

**Review of the article "Dense water formation in the Eastern Mediterranean under global warming scenario" by Parras-Berrocal et al.**

**In this paper the authors study the evolution of the dense water formation in the Eastern Mediterranean (EMED) along the 21st century, under RCP 8.5 greenhouse gases emission scenario, using the ROM Regional Climate Model (RCM). This RCM have been previously validated and used in several climate studies for the Mediterranean basin and sub-basins. They find a significant reduction of the deep water formation (DWF), between 75% and 85%, in the three regions were deep and intermediate convection take place (Adriatic, Aegean and Levantine basin) by the end of the century. The authors identify the increase in the water column stratification, due to the projected warming and salinization caused by the global warming, as the main factor driving this DWF reduction. They also predict a shift in the main Eastern Mediterranean Deep Water (EMDW) formation region, from the Adriatic to the Aegean Sea, similar to what occurred in the mid-90s with the so called Eastern Mediterranean Transient (EMT).**

**I find the paper very interesting. As the authors mention, this topic has not been studied in depth and it is very important to understand the expected changes in the Mediterranean thermohaline circulation as a consequence of the climate change. Although the use of a single model limits the robustness of the results, as the authors themselves point out, the results presented and the analysis of the mechanisms behind them are very relevant to the climate modeling community of the Mediterranean region. The manuscript is well written and organized, all the ideas are concisely and clearly stated, and well referenced. There are only a couple aspects that, in my opinion, need to be clarified before its publication in OS.**

**The first and more important one is the selection of the density of reference used to estimate the DWF rate in the model. As the authors themselves explain, the potential density they use in the model for the newly formed deep water in the Adriatic, Aegean and Levantine basins is slightly lower than the observed and reported in the literature. This is due to the lower salinity, and hence density, of the model respect to the observations in these regions (figure S1). Adjusting the reference density to the 'model reality' is a sound methodology, so this shouldn't be a problem. It would be interesting, though, to identify these references (both for observations and model) in the profiles of figure S1. The selected reference corresponds with approximately 650 m depth in the historical period. However, the authors maintain the same reference densities in the future to compute the DWF rate evolution, which could have led to an underestimation. The profiles for the projections seem to show a general reduction of the density in the whole water column for the Adriatic and the Aegean regions (figures S2,3). This mean that the density of reference would correspond with a deeper layer, and that future deep water might be lighter than the present one. It is difficult to identify these differences in the figures, and very likely there will be no significant variations in the results, but in my opinion the authors should clarify this point in in the results or discussion sections. Would the DWF rate increase if a different density of reference is used for the future? Maybe including a third set of panels with the evolution of the density in figure 4 would be also of help.**

**Response:** Thank you for the remark. We have identified the density of reference in the profiles of ROM and WOA18 (Figure S1):

[Figure]

**Figure S1. Winter spatially and temporally averaged vertical profiles of (a, d, g) temperature, (b, e, h) salinity (psu) and (c, f, i) density (kg/m³) in the Adriatic Sea (top), Aegean Sea (middle), and Levantine Sea (bottom) for the 1980-2012 period. ROM_P0 (blue) and WOA18 (red). Note different ranges in horizontal axes.**

We estimate the future reduction of dense water formation by using hydrographic properties of the present climate (ROM_P0) as a reference. However, we acknowledge the concerns raised by Reviewer regarding this approach. Indeed addressing that concern is a challenge, as the density vertical structure is changing differently in different sub-basins, so the "present" deep water density thresholds might not be applicable to sub-basins with strong changes in density vertical structure in the future. On the other hand, the water formation rate above a certain density threshold is a model output, which allows a direct comparison between "present" and "future", but as the Reviewer points out the issue of a future lighter sea must be taken with care and properly addressed in the discussion. Our projections indicate that changes in the hydrographic properties of the water column are likely to lead to a reduction in density, which may affect our estimates. In order to address this comment, we have computed the water column density anomalies over the Adriatic, Aegean, and Levantine Seas (as shown in Figure 1). These anomalies were defined as the difference between the averaged vertical profile of density for the ROM_P2 period (2070-2099) and the present climate period (ROM_P1, 1976-2005). By comparing these density anomalies, we were able to identify changes in the hydrographic properties of the water column over time and across different geographic regions.

By the end of the century, the Adriatic Sea experienced a decrease in density of 0.1 kg/m³ at a depth of 650 m, while the average decrease over the entire water column was 0.2 kg/m³ (Figure 1d). In contrast, in the Aegean Sea we found no changes in density at the 650 m depth, and the averaged over the entire water column decrease was 0.08 kg/m³ (Figure 1e). In the Levantine Sea, we observed a reduction in density of 0.15 kg/m³ at both 300 m depth and throughout the entire water column (Figure 1f).

[Figure]

**Figure 1.** Spatially and temporally averaged vertical profiles of density in the (a) Adriatic, (b) Aegean, and (c) Levantine Seas values corresponding to the historical period (1976-2005), future projection (2070-2099). (d, e, f) The difference between future and historical averaged vertical profiles are also shown.

Based on the observed density reductions (Figure 1), we calculated the potential changes in dense water formation rates in the Adriatic, Aegean, and Levantine Seas (Figure 2). In the Adriatic Sea, using a density reference of 29 kg/m$^3$, we projected a 75% reduction in DWF rate. Using lower density references of 28.9 kg/m$^3$ and 28.8 kg/m$^3$ resulted in smaller reductions of 58% and 39%, respectively. In the Aegean Sea, where the density is expected to remain relatively stable, the differences in the calculated changes in DWF rate are negligible, with a projected decrease that varies from 84% (using a density of 28.95 kg/m$^3$) to 80% (using 29.9 kg/m$^3$). In the Levantine Sea, the reduction of the projected decrease in DWF rate is more substantial, with a drop from 83% (using a density of 28.7 kg/m$^3$) to 56% (using 29.55 kg/m$^3$).

[Figure]

**Figure 2.** Time series (1976–2099) of ROM_P1 and ROM_P2 simulations of yearly DWF rate (Sv) averaged over (a) Adriatic Sea, (b) Aegean Sea and (c) Levantine Sea. d) Averaged DWF rates for 1976-2005 (triangle) and 2070-2099 (circle) periods.

Overall, the use of a lower isopycnal threshold leads to the same conclusions: a collapse of DWF in the Aegean Sea (Figure 2b) and a notable reduction (of more than 50%) in the Levantine basin (Figure 2c).

However, in the Adriatic Sea this is not the case: when using the lowest isopycnal density threshold (28.8 kg/m$^3$, blue line in Figure 2a) there are events with notable DWF rates during the last third of the century.

After conducting a deeper analysis, as suggested by the reviewer, we noted different DWF rate responses to changes in density threshold for each sub-basin. In the Adriatic Sea, the future changes on DWF strongly depends on the density change in the water column and therefore on the choice of the density threshold, with denser isopycnals corresponding to stronger DWF rate reductions, with no remarkable DWF change for the less dense threshold (Figure 2d). In the Aegean Sea, a collapse of DWF is expected regardless of the density reference used. Finally, in the Levantine Sea, the changes in DWF rate appear to have a limited dependence on the choice of the density threshold. The reduction in the DWF rate is 83% for 28.7 kg/m$^3$ and 56% for 28.55 kg/m$^3$, so the reduction is noticeable in both cases, although sensitive to the choice of the density threshold. We will address this point in the discussion of the revised manuscript.

We have included a third set of panels with the evolution of the density in Figure 4 of the manuscript:

[Figure]

**Figure 3. ROM RCP8.5 time series (1976-2099) of 0-1000 m (a, b, c) potential temperature (ºC), (d, e, f) salinity (psu) and (g, h, i) potential density (kg/m³). All-time series correspond to winter months (December-January-February-March) in the Adriatic (upper row), Aegean (middle row) and Levantine (bottom row) Seas.**

**My second concern is that the authors did not describe the limits of the regions of each sub-basin used to estimate the average profiles shown in the figures and the DWF rates. As they point out in the introduction, the areas of the Aegean and Adriatic where the deep convection take place are very specific. Are the profiles and DWF rates estimated in these specific areas or in the whole sub-basins? When computing the volume of deep water formed every year, do they account for all the volume of water with densities higher than the reference in a specific region, in the whole sub-basin or in the whole EMED (maybe considering the possible spread)? The region selected could also modify the results, so I think this should also be clarified in the MS. Maybe you could include the basins limits in figure 1 (the color scale is also missing).**

Response: Thank you for the comment. We have estimated the DWF rates and profiles considering the whole sub-basins (Adriatic, Aegean and Levantine).

We computed the volume of the deep water formed every year in the whole sub-basins, so we could take into account the non-negligible amount of deep water formed on the continental shelf and subsequently spread in the sub-basin (Lascaratos et al., 1999, Nittis et al., 2003) and to make our model estimates comparable to other estimates existing in the literature (Nittis et al., 2003; Mantziafou and Lascaratos 2008). ROM_P0 has shown a good representation of the deep water volume formed during present climate. In

order to clearly show the limits that define each sub-basin we have modified Figure 1 as follows:

[Figure]

**Figure 1. a) Bathymetry of the main spots for dense water formation in the EMed: Adriatic Sea, Aegean Sea and Levantine Sea. The main water masses of each spot sorted by depth range are also shown: [0-100 m] Adriatic Surface Water (AdSW), Black Sea Water (BSW), Levantine Surface Water (LSW); [100-650 m] Levantine Intermediate Water (LIW), Cretan Intermediate Water (CIW); [650-1000 m] Adriatic Deep Water (ADW), Cretan Deep Water (CDW) and Levantine Deep Water (LDW). b) The domain used for the calculations in each sub-basin are colored: orange-Adriatic, blue-Aegean and green-Levantine.**

---

## Author Comment (AC2)

**Response to Reviewer #2**

We thank the Reviewer #2 for the effort in reviewing the manuscript and for her/his positive evaluation. The posted comments have helped us to improve the manuscript and make it more robust and complete.

**Reviewer #2 (Comments to Author (shown to authors):**

**The authors investigated the variations of dense water formation (DWF) in the Eastern Mediterranean (EMed) through the twenty-first century under the RCP8.5 emission scenario for understanding the impacts of climate changes on the Mediterranean overturning circulation. Their results indicated that the dominant source of Eastern Mediterranean Deep Water (EMDM) shifts from the Adriatic Sea to the Aegean Sea during the 2005-2040 period. By the end of the century, DWF for the Adriatic Sea, the Aegean Sean, and the Levantine Sea all perform a pronounced decrease by 75%, 84%, and 83%, respectively, which is a result of hydrographic changes of surface and intermediate water and the associated strengthening water column stratification under the RCP8.5 emission scenario. The results shown are impressive and, as was pointed out in the manuscript, also fill in the gap of the DWF study in the EMed providing a more quantitative assessment than previous studies. The manuscript was also well-written and easy to follow. But some improvements may be needed before the publication.**

**1) The coverages of the Adriatic Sea, the Aegean, and the Levantine Sea should be specified and shown in a figure as results and discussions of this study focus on the DWF from these regions. Thus, it is important to provide the spatial extent of these basins, which can also help readers to understand the studied area better. In addition, as it was stated that the horizontal resolution of the model varies from 7 km to 25 km (which is a big difference, I think), it is better to show the computational grid as well.**

**Response:** Thank you for the comment. In order to represent the spatial extent of each sub-basin and the oceanic computational grid, we have modified Figure 1 as follows:

[Figure]

**Figure 1.** a) Oceanic computational grid and resulution adopted in ROM (in km, only one out of four lines are drawn). The domain used for the calculations in each sub-basin are souounded by color lines: Adriatic (red), Aegean (orange)

and Levantine (grey). b) Bathymetry of the main spots for dense water formation in the EMed: Adriatic Sea, Aegean Sea and Levantine Sea. The main water masses of each spot sorted by depth range are also shown: [0-100 m] Adriatic Surface Water (AdSW), Black Sea Water (BSW), Levantine Surface Water (LSW); [100-650 m] Levantine Intermediate Water (LIW), Cretan Intermediate Water (CIW); [650-1000 m] Adriatic Deep Water (ADW), Cretan Deep Water (CDW) and Levantine Deep Water (LDW).

**2) Statistics analysis and parameters are needed. Firstly, the authors may need to provide p values for every correlation coefficient as they are important to illustrate the significance. Secondly, the 2040s was regarded as a time point around which sharp changes in SI and DWF (Figure 3) were observed. However, the author may need to provide a more convincing way to address this time point not just by the naked eye but using some statistical tools, like the non-parametric change-point Pettitt test (Pettitt, A.N. A non-parametric approach to the change-point problem. Appl. Stat. 1979, 28, 126–135).**

**Response:** We agree with the reviewer. Therefore, we have tested every correlation coefficients at the 95% confidence level and the p-values obtained have been included in the revised manuscript. Now reads:

Line 111f: "The interannual DWF rate in the Adriatic Sea (Figure 2a) agrees well (r=0.70 at 95% of confidence level, p-value=0.001) with estimates based on the Princeton Ocean Model (POM) of Mantziafou and Lascaratos (2008)."

Line 116f: "On the other hand, the interannual DWF rate in the Aegean Sea (Figure 2b) is also well correlated (r=0.75, p-value=0.002) with the POM results reported by Nittis et al. (2003)."

Line 171f: "Our results indicates that the intensity of DWF rate is mostly determined by the SI (Pearson correlation coefficient (r) > 0.7 and p-values=0 in all regions), as low or high amount of water produced can be found with similar BL values (r < 0.1 and p-values > 0.05) (Figure 3)."

P-values for DWF rate vs. BL: Adriatic Sea (0.55), Aegean Sea (0.13) and Levantine Sea (0.78).

Following the indications of the reviewer, we have applied the non-parametric change-point Pettitt test in order to provide convincing result of abrupt shifts in SI and DWF rates.

Adriatic Sea: DWF rate (year = 2054, K= 2.937, p-value=0) | SI (year = 2039, K= 3.104, p-value=0)

Aegean Sea: DWF rate (year = 2040, K= 2.457, p-value=0) | SI (year = 2040, K= 3.639, p-value=0)

Levantine Sea: DWF rate (year = 2054, K= 2.918, p-value=0) | SI (year = 2041, K= 3.764, p-value=0)

Abrupt changes in DWF rates are projected to occur by 2054 in Adriatic and Levantine Seas while in the Aegean the shift happens by 2040. The expected SI changes in all regions take place around 2040. We will include these results in the revised manuscript.

**3) Could you double-check the unit "Sv yr" which first appears on Line 113? If my understanding of the unit "Sv" is correct, Line 113 should be rewritten as:**

*During 1981-1999, ROM_P0 produces a total of 5.45 Sv of newly waters denser than 29.0 kg/m3 corresponding to an annual formation rate of 0.29 Sv…*

**Response:** We are sorry for the confusion. We use "Sv yr" because we do not consider the DWF rate but the total volume of dense water formed, following Nittis et al. (2003).

**4) Lines 202-211. Although the authors provided descriptions of SI for different periods, I am still not quite sure how the authors calculated the percentage contributions of different water bodies to the temporal changes in SI. Could you please provide some descriptions or equations to further address the calculation?**

**Response:** We calculate the relative contributions of different water masses using the SI number of Figures S2, S3 and S4, as previously carried out in Parras-Berrocal et al. (2022).

We use data from Aegean Sea as example (see Table 1 of the manuscript):

The total SI change (Proj. – Hist.: 2.13-1.40 = 0.73 $m^2s^{-2}$) and the BSL/LSW layer accounts for (2.13-1.54= 0.59 $m^2s^{-2}$) whereas the CIW/LIW layer accounts for (2.13-1.99= 0.14 $m^2s^{-2}$).

Then, we applied a simple rule of three:

$$BSL/LSW: (((0.73-0.59)*100)/0.73) = 19.2 \%$$
$$CIW/LIW: (((0.73-0.14)*100)/0.73) = 80.8 \%$$

Thanks to the Referee comment we have detected an error/typo. We apologize for the error in lines 209-211, where the percentages of each contribution were mistakenly interchanged. We have changed it in the revised manuscript and now reads:

"In the Aegean Sea, the alteration in BSW/LSW properties leads the 19.2% of the total SI future change while the change in CIW/LIW is 80.8%. Finally, in the Levantine Sea changes in the MAW/LSW properties contributes in a 40% whereas LIW in a 60%."

**5) Lines 268-270. It may be a jump to conclude that the increasing potential density is caused by the increasing salinity over the upper 100 m, as the authors only compared the salinity changes and density changes (Figure S7) but ignored the contributions of temperature changes. As shown in Figure 4 subsurface (0-100 m) temperature seemly performs an increase in the ADR (Figure 4a) from the period of 2006-2020 to the period of 2020-2040 but fluctuates in the AEG (Figure 4c) and LEV (Figure 4e). Thus, the author may need to quantify both contributions of the changes in temperature and salinity to the changes in density.**

**Response:** Thank you for the remark. In order to address this comment, we have evaluated the spatially and temporally averaged vertical profiles (0-100 m depth) of temperature, salinity, and density for the 2006-2020 (solid lines) and 2020-2040 (dashed lines) periods in the Adriatic, Aegean, and Levantine Seas (Figure R1). In the bottom row, we also show two synthetic density profiles: (red line) displays the density profile keeping the salinity fixed (2006-2020) whereas the temperature is evolving (2020-2040) and (blue line) the opposite.

[Figure]

**Figure R1.** Spatially and temporally averaged vertical profiles (0-100 m depth) of temperature, salinity, and density for the 2006-2020 (present (p), solid lines) and 2020-2040 (future (f), dashed lines) periods in the Adriatic, Aegean, and Levantine Seas. (d, e, f) It is shown two synthetic density profiles: (red line) display the density profile keeping the salinity fixed (2006-2020) whereas the temperature is evolving (2020-2040) and (blue line) the opposite.

**Table R1.** Quantification of the relative contributions of temperature and salinity to changes in density calculate from vertical profiles presented in Figure R1.

| | Adriatic Sea | | Aegean Sea | | Levantine Sea | |
|---|---|---|---|---|---|---|
| | $\rho$ (kg/m³) | % of each contribution | $\rho$ (kg/m³) | % of each contribution | $\rho$ (kg/m³) | % of each contribution |
| $T_p$-$S_p$ (2006-2020) | 28.76 | - | 28.23 | - | 28.38 | - |
| $T_f$-$S_f$ (2020-2040) | 28.94 | | 28.42 | - | 28.49 | - |
| $T_f$-$S_p$ | 28.76 | 0% | 28.29 | 31.6% | 28.42 | 36.4% |
| $T_p$-$S_f$ | 28.94 | 100% | 28.36 | 68.4% | 28.45 | 63.6% |

In the Adriatic Sea, changes in salinity are responsible of nearly the total density change expected in the 2020-2040 period (Table R1). In the Aegean and Levantine Sea, changes in salinity contribution to density changes (2020-2040) are 68.4% and 63.6%, respectively (Table R1). These results suggest that salinity changes are the only cause of those density changes in the Adriatic Sea and a primary factor in the Aegean and Levantine Seas. We will address this point in the revised manuscript.

**6) Lines 270-271. The authors may need to provide more evidence in addressing the causes of the changes in the upper ocean circulation, like correlations between changes in salinity or temperature and changes in circulation patterns. Or to provide some mechanistic explanations on how the changes in salinity or temperature would lead to changes in circulation. Or to provide results of previous studies here that may have such discussions.**

**Response:** Thank you for your comment. We agree with the reviewer. In lines 270-271, we advanced a suggestion that "the increase of the salt content in the 0-100 m layer could be the result of a change in the upper ocean circulation that alters the AW path", which is clearly a hypothesis that needs further in-depth work and it is not properly placed. We understand that this further analysis is out of the scope of this work therefore we have decided to remove Lines 270f and the Figure S7 from the revised manuscript.

However, in order to provide an extended response to this comment we explored some of the possibilities indicated by the Reviewer, and we found in previous works (Gasparini et al., 2005; Incarbona et al., 2016) that in periods of intensified EMed DWF, for example during the EMT, there is a decrease in the salinity in the AW in the Sicily channel, which seems to be provoked by an enhancement in the MTHC which implies an increase in the energy inflow at the Strait of Gibraltar. Such more energetic inflow results in the AW more directly conveyed to the Sicily Channel, therefore less mixed with the surrounding resident water, resulting in a decrease of salt transport to the EMed in the upper layer through the Sicily Channel.

In fact in our results, a significant part of the period 2020-2040 (period with the higher DWF rates, Figure 3 of the manuscript) we found a negative upper layer salt transport anomaly through the Sicily Channel towards the Eastern basin. We have calculated the anomalies of salt transport through the Sicily Channel (Figure R3) by using a moving mean filter to reduce the noise signal, and then subtracting the trend of the time series. We detected that there is a certain significant negative correlation (r=-0.38, p-value=0) between the upper salt transport through the Sicily Channel and the 0-100 m salinity of EMed (Figure R3b). While very interesting, the results are not conclusive in our opinion, and demand complimentary work.

[Figure]

**Figure R2.** a) Transect in with the transport across the Sicily Channel is calculated. b) Time series (2006–2050) of ROM_P2 simulation of yearly [red] salt transport anomaly through the Sicily Channel integrated for the 0-100 m depth and [blue] salinity anomaly (psu) averaged for the layers 0–100 m in the EMed.

As we stated above, we are very grateful for her/his critical remark. This is an interesting issue, but it requires a targeted and more in-depth analysis, which could be addressed in forthcoming studies.

**References:**

Gasparini, G.P., Ortona, A., Budillon, G., Astraldi, M., and Sansone, E. (2005). The effect of the Eastern Mediterranean Transient on the hydrographic characteristics in the Strait of Sicily and in the Tyrrhenian Sea. Deep-Sea Res. I, 53, 915-935, doi: 10:1016/j.dsr.2005.01.001

Incarbona, A., Martrat, B., Mortyn, P.G., Sprovieri, M., Ziveri, P., Gogou, A., Jordà, G., Xoplaki, E., Luterbacher, J., Langone, L., Marino, G., Rodríguez-Sanz, L., Triantaphyllou, M., Di Stefano, E., Grimalt, J-O., Tranchida, G., Sprovieri, R., and Mazzola, S.: Mediterranean circulation perturbations over the last five centuries: Relevance to past Eastern Mediterranean transient-type events, Sci. Rep-UK, 6, 1-10, doi:10.1038/srep29623, 2016.